# Lack of Prophylactic Cranial Irradiation for Extensive Small-Cell Lung Cancer in Real Life, with the Emergence of Immunotherapy

**DOI:** 10.3390/cancers16234122

**Published:** 2024-12-09

**Authors:** Alice Daumas, Celestin Bigarre, Mohamed Boucekine, Audrey Zaccariotto, Bertrand Kaeppelin, Alice Mogenet, Etienne Gouton, Johan Pluvy, Pascale Tomasini, Xavier Muracciole, Sebastien Benzekry, Laurent Greillier, Laetitia Padovani

**Affiliations:** 1Oncology Radiotherapy Department, Assistance Publique des Hôpitaux de Marseille, Aix-Marseille Université, 13005 Marseille, France; alice-julie.daumas@ap-hm.fr (A.D.);; 2COMPO, Inria Méditerranée, Cancer Research Center of Marseille, Inserm UMR1068, CNRS UMR7258, UM105, Aix-Marseille Université, 13273 Marseille, France; 3Unity of Research EA3279, Aix-Marseille Université, 13007 Marseille, France; 4Multidisciplinary Oncology and Therapeutic Innovations Department, Assistance Publique des Hôpitaux de Marseille, Aix-Marseille Université, 13005 Marseille, France; alice.mogenet@ap-hm.fr (A.M.); johan.pluvy@ap-hm.fr (J.P.);; 5Scientific Research National Center (CNRS), Institute of Neurophysiopathology, Aix-Marseille University, 13005 Marseille, France

**Keywords:** extensive-stage small-cell lung cancer, prophylactic cranial irradiation, immunotherapy

## Abstract

Prophylactic cranial irradiation (PCI) is recommended to decrease the incidence of brain metastases (BM) in extensive stage small cell lung cancer (ESSCLC) without BM after response to chemotherapy while it is known also to be associated with significant neurocognitive effects. In recent years, advancements in radiotherapy technologies and the availability of MRI have led to stereotactic radiotherapy being a preferred option for brain metastases in various primary cancers, including non-small cell lung cancer (NSCLC). Several studies about BM in NSCLC have reported intracranial activity signal and significant improvement of intracerebral disease control rates with immunotherapy treatment However the brain effetc of immunothrapy remains unclear. We report a single-center retrospective study evaluating the impact of omitting. PCI from real-life treatment, including immunotherapy, of patients with ES-SCLC. About a cohort of 56 patients, A recursive partitioning analysis (RPA) found PS, immunotherapy and age to be influential factors for OS but not PCI.

## 1. Introduction

Small-cell lung cancer (SCLC) accounts for approximately 15% of all lung cancers. During follow-up, brain metastases (BM) occur in 70 to 80% of cases, resulting in a poor prognosis with a 5-year survival rate of 15–25% regardless of the initial stage [1,2]. The median survival for extensive-stage SCLC (ES-SCLC) is around 7 to 11 months. Until recently, the standard of care was a systemic treatment based on a combination of platinum-based chemotherapy [3]. This treatment was followed by prophylactic cranial irradiation (PCI) for patients with a partial or complete response without BM. These recommendations for PCI are based on the findings of two meta-analyses, which reported a benefit of PCI on overall survival (OS) compared to no irradiation after complete response to radio chemotherapy for limited or chemotherapy for disseminated SCLC [4,5]. These results were further supported by a randomized phase 3 EORTC trial published by Slotman et al. [6] in 2007. The study focused on ES-SCLC patients with a partial or complete response and reported a significant increase in both brain metastasis-free survival (BMFS) and OS (hazard ratio (HR) of 0.68 (IC95% [0.52; 0.88] for OS) in the PCI group. Nevertheless, patients did not undergo brain magnetic resonance imaging (MRI) before radiotherapy, so the evaluation of cerebral disease before PCI may have been underestimated [7]. In addition, in 2017, Takahashi et al. [8] reported the results of a study comparing PCI with an MRI follow-up every 3 months vs. an MRI follow-up only in patients with ES-SCLC. The study found no statistically significant difference in OS (13.7 months vs. 11.6 months, HR = 1.27 (CI95% [0.96; 1.68]) between the two populations with an increase in side effects in the PCI group. Despite this trial, European recommendations still support the use of PCI for patients under 75 years old, with a performance status (PS) from 0 to 2, with no progression after chemotherapy [3]. A recently published meta-analysis found that PCI was associated with improved OS and a reduced cumulative incidence of BM in patients with SCLC, regardless of disease stage. But in a subgroup analysis limited to studies in which all patients underwent MRI, excluding patients with BM, PCI was associated with a reduced incidence of BM without an effect on OS [9]. From different adult and pediatric studies, it is known that whole-brain irradiation induces major neurocognitive side effects [10,11,12]. In recent years, advancements in radiotherapy technologies and the availability of MRI have led to stereotactic radiotherapy being a preferred option for brain metastases in various primary cancers, including non-small-cell lung cancer (NSCLC). According to the ASTRO 2021 recommendations [13], whole-brain radiotherapy (WBRT) has largely been replaced by focal treatment whenever possible, without compromising OS [14,15,16] in other cancers. In BM of SCLC, a retrospective study published in 2020 found no difference in OS between radiosurgery and WBRT [17], and a phase 3 trial (NRG CC009) is currently recruiting to answer this question.

Furthermore, the emergence of immunotherapy against SCLC [18,19] with controversial effects on BM raises questions about the role of PCI. Several studies about BM in NSCLC have reported intracranial activity signal and a significant improvement in intracerebral disease control rates with immunotherapy treatment [20,21,22]. Since the ESMO 2021 recommendations [3], the new standard of care in ES-SCLC is a combination of immunotherapy and chemotherapy using atezolizumab or durvalumab in concomitant and maintenance treatment, but there are very limited data on immunotherapy and PCI. The IM-Power 133 study showed that the monoclonal antibody atezolizumab significantly delayed the time to intracranial progression (20.2 months vs. 10.5 months, *p* = 0.046) compared to a placebo. Although the PCI was allowed in both groups in this trial, only a few patients actually received it [18]. Likewise, in the CASPIAN Trial [19], PCI was allowed only in the standard chemotherapy arm, and only 21 patients out of 269 received it.

In this context of lack of data including immunotherapy in the treatment schedule, we report here a single-center retrospective study evaluating the impact of omitting PCI from real-life treatment, including immunotherapy, on OS and BMFS in patients with ES-SCLC, over the period January 2014 to January 2021.

## 2. Materials and Methods

Population: This is a retrospective, single-center study of patients treated for ES-SCLC with chemotherapy, at Assistance Publique des Hôpitaux de Marseille (APHM) between January 2014 and January 2021, with an indication of PCI after partial response or stable disease. Inclusion criteria were histologically proven ES-SCLC, PS ≤ 2, age > 18 years, diagnosis between January 2014 and January 2021, and follow-up at APHM. Non-inclusion criteria were the presence of initial brain metastases, and extra-cerebral or cerebral progression either ongoing or immediately after the first treatment, PS ≥ 3. Initial patient characteristics, details of initial treatments, and clinical data were extracted from medical records. The date of diagnosis was the date of histological sampling. The date of last treatment was the date of the last cycle of chemotherapy. Chemotherapy treatment was platinum- and etoposide-based. A reevaluation by a CT scan of chest/abdomen/pelvis and brain imaging (CT or MRI) was scheduled less than 2 months after the last treatment date. Complete or partial response and stability were defined according to the Response Evaluation Criteria in Solid Tumors (RECIST) version 1.1. The performance of PCI was modulated according to the patient’s neurocognitive status and was not performed if the patient refused. PCI was delivered using linear accelerators with a mean total dose of 24 Gy. All patients signed an informed consent form. The data used were anonymized, collected from the APHM computer file, and declared to the APHM health data access portal. The study received approval from the Institutional Review Board of the French learned society for respiratory medicine, Société de Pneumologie de Langue Française.

The primary endpoint of the study was OS, while secondary endpoints were BMFS, extra- cerebral metastasis-free survival (ECMFS), and survival without a second cerebral event. A cerebral event was defined as the appearance or progression of one or more brain metastases. The first cerebral event was defined as the date of the first imaging (CT or MRI) showing a suspicious brain lesion. Overall, survival was defined as the time between last treatment and death from any cause, censored at the date of last follow-up. BMFS and ECMFS were defined by the time between the date of inclusion (date of last treatment) and the occurrence of the respective specific events, censored at the date of last follow-up.

Statistical analysis: Descriptive analysis was carried out to compare the main characteristics of the overall population. Patient and treatment characteristics were presented as numbers and percentages for categorical variables, and as means ± standard deviation for continuous variables. The significance of differences was determined using the chi-square test or Fisher’s exact test for categorical variables and using the *t*-test or Mann–Whitney test for continuous variables, as appropriate. To determine OS, BMFS, and ECMFS, we used the Kaplan–Meier method to estimate median survival with a 95 percent confidence interval and the Cox model to determine hazard ratios (HR) for comparisons between groups (PCI vs. no-PCI) with a 95% confidence interval. Univariate and multivariate Cox analyses were used to identify prognostic factors for OS and BMFS. The multivariate analysis included all univariate significant clinical variables (*p* < 0.2) according to the performance or not of initial PCI. *p* < 0.05 was considered significant. The 95% confidence interval was reported. Statistical analysis was performed using IBM SPSS Statistics for Windows, version 21.0 (IBM SPSS Inc., Chicago, IL, USA). As an exploratory analysis, a recursive partitioning analysis (RPA) using a survival tree for OS was generated based upon the most statistically significant variables in the univariate models and additional clinically meaningful variables. RPA divided patients at each step into two groups based on the covariate that provided maximum separation with respect to prognosis and accounted for interactions between factors. The RPA algorithm is based on the optimized binary partition of these subgroups, which enables the classification of patients into successively more homogeneous prognostic groups based on multiple input variables. RPA performs a ten-fold cross-classification by default to help assess the reliability of the tree model. The full sample is randomly divided into 10 sub-samples. Internally, the full RPA tree is carried out with 90% of the full sample, and the remaining 10% of the sample is used as a validation dataset to calculate a cross-classification error rate. This procedure is repeated 10 times, each time with 9 subsets as the modeling dataset and the remaining 1 subset as the validation dataset. Additional technical details can be found in this document [23].

## 3. Results

A total of 287 patients were treated for SCLC between 2014 and 2021 at APHM. Among them, 79 initially had BM. One hundred one either developed BM or extra-cranial metastases before the date for PCI or died during chemotherapy. A total of 107 patients were eligible to receive PCI, of whom 56 were initially diagnosed with ES-SCLC and were included in our study. Among these patients, 25 received PCI, 31 had no PCI, and 18 received immunotherapy. The initial characteristics of the population are described in Table 1.

Median follow-up was 16 months for the whole population. There was no significant difference in terms of BMFS (*p* = 0.336) and OS (*p* = 0.412) between the two groups Figure 1. Median OS and BMFS were, respectively, 11.7 and 13.4 months in patients with PCI, and 20.3 and 10.7 months in patients without PCI. Median ECMFS was also comparable in the two groups, with 4.6 and 4.2 months, respectively, in the no-PCI and PCI groups (*p* = 0.446).

At the first cerebral event, the PS, size, and number of brain metastases were not statistically different between the two groups. PCI had no significant impact on the occurrence of a second cerebral event either from the inclusion date (*p* = 0.77) or from the first event (*p* = 0.669). Because the two groups were not balanced regarding immunotherapy, we also compared patients receiving immunotherapy or not in terms of survival. There was a trend in favor of immunotherapy but it was still not significant, in terms of OS (11.6 vs. 23.6 months *p* = 0.054) and ECMFS (4 vs. 5.7 months, *p* = 0.052). We observed a trend towards longer BMFS in immunotherapy-treated patients Figure 2, which, however, did not reach statistical significance.

We performed univariate and multivariate analysis to determine the prognostic factors for OS and BMFS, including the PCI and adjuvant immunotherapy status. Univariate and multivariate analysis are summarized in Table 2 and Table 3. In multivariate analysis, only the ratio MLR was found to be associated with BMFS (*p* = 0.034), and PS ≥ 2 was found to be associated with OS (*p* = 0.014).

We performed a RPA to identify predictors of survival. The RPA divided patients into four risk groups according to PS, immunotherapy treatment, and age at treatment but not PCI (Figure 3).

## 4. Discussion

ESMO 2021 recommendations are still in favor of performing PCI on patients with no progressive disease [3] after chemotherapy and remain unclear about PCI for patients treated with immunotherapy. In this context, the current study is one of the rare real-life studies reporting the results of omitting PCI for ES-SCLC integrating immunotherapy in the first-line strategy. We did not find that PCI had a positive impact in terms of OS and BMFS in our study population. Our data were in contradiction with the 2007 EORTC trial [6], which reported a benefit with PCI in terms of OS and BMFS. Moreover, OS was twice as long in both of our groups compared to this trial. In our study, median OS was 11.7 months for patients who received PCI and 20.3 months for those who did not, compared to only 6.7 months and 5.4 months, respectively, in the 2007 trial. This difference can be partly explained by poorer patient selection, especially in the absence of pre-therapeutic MRI, but mainly by the absence of immunotherapy at that time, which contributed to improving the OS of our patients.

However, our results in terms of OS remained consistent with more recent literature data. In a Japanese study [8] of ES-SCLC, median survival was comparable, with 11.6 months in the PCI group and 13.7 months in the MRI group, vs. 11.7 months and 20.3 months, respectively, in our study. Nevertheless, in their trial, PCI had an impact on reducing the cumulative incidence of BM, which was not observed in our study. It is important to note that the primary endpoint was OS for their trial, but it was stopped early at a planned interim analysis due to the demonstration of the futility of PCI.

The introduction of immunotherapy during this study period can explain the lack of impact of PCI on BMFS. The introduction of immunotherapy for patients after positive phase III trials has changed medical practices, and for these patients, PCI was more frequently omitted, as allowed in the trials guidelines [18,19]. In summary, our results could be interpreted either by the introduction of immunotherapy or by a lack of efficiency of PCI, supporting the idea that immunotherapy has modified the natural course of brain disease. Furthermore, in our RPA results, immunotherapy stands out as the second most important factor affecting overall survival, following PS, while PCI was not established as a relevant factor. These results support the impact of immunotherapy and underline the lack of PCI efficiency.

The PACIFIC trial showed that adjuvant therapy with the human monoclonal antibody durvalumab reduced the incidence of BM from 11.8% to 6.3% [24] in NSCLC, as was observed in the IM-POWER trial for SCLC and other trials [25]. Thus, the use of immunotherapy may potentially negate the marginal OS benefit of adding PCI.

The relationship between immunotherapy and brain control is still debated in the literature. No available randomized trials can help make the decision to defer brain irradiation in patients receiving immunotherapy. In this context, ESMO 2021 recommendations [3] still support PCI in progression-free patients without BM after chemotherapy who are under 75 years old and have a PS ≤ 2. However, they agree on the lack of appropriate data in patients undergoing immunotherapy and propose to share decision making for these patients. Meanwhile, NCCN currently recommends shared decision making regarding PCI for patients with ES-SCLC with a response to initial treatment with or without immunotherapy [26].

In our study, we observed that the MLR ratio had a potential impact on BMFS. A lower MLR (<0.12) was associated with an increased risk of cerebral progression with an HR of 1.21 (1.01–1.45). These data are comparable with the existing literature, particularly with the use of this ratio in nomograms attempting to predict the occurrence of BM in limited stage SCLC [27,28]. In contrast, other studies find a protective effect of a low MLR ratio in NSCLC [29,30] and in all other histologies [31]. Further studies are needed to validate the real role of the MLR ratio. However, the use of a biomarker such as MLR could help to predict the risk of cerebral progression and then to select high-risk patients who could benefit from PCI.

The limitations of our study primarily concern its retrospective nature and the small size of our series. The difference between the two groups regarding immunotherapy is also a limitation that prevents us from drawing definitive conclusions about the absence of the benefit of PCI on BMFS or the possible benefit of immunotherapy. Furthermore, due to ESMO 2021 recommendations, exhaustive monitoring via brain MRI was not systematically performed [3]. However, this study represents one of the rare clinical investigations reporting data on PCI in the context of immunotherapy for ES-SCLC and reports a strong signal in favor of omitting PCI. The results of Primalung (NCT0479025) and Swog-1827 are eagerly awaited.

## 5. Conclusions

In conclusion, the clinical results of our study do not show any benefit of PCI in terms of OS and BMFS for patients with ES-SCLC. Considering the current systemic strategies, a therapeutic de-escalation should be assessed in randomized trials that incorporate immunotherapy.

## Figures and Tables

**Figure 1 cancers-16-04122-f001:**
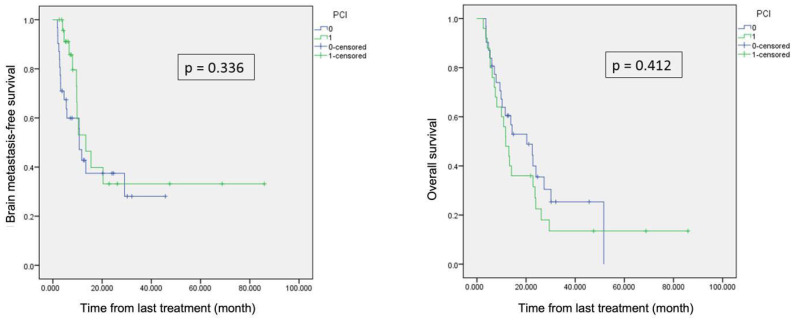
(**left**) Brain metastasis-free survival; (**right**) overall survival. Patients with PCI in green, and patients without PCI in blue.

**Figure 2 cancers-16-04122-f002:**
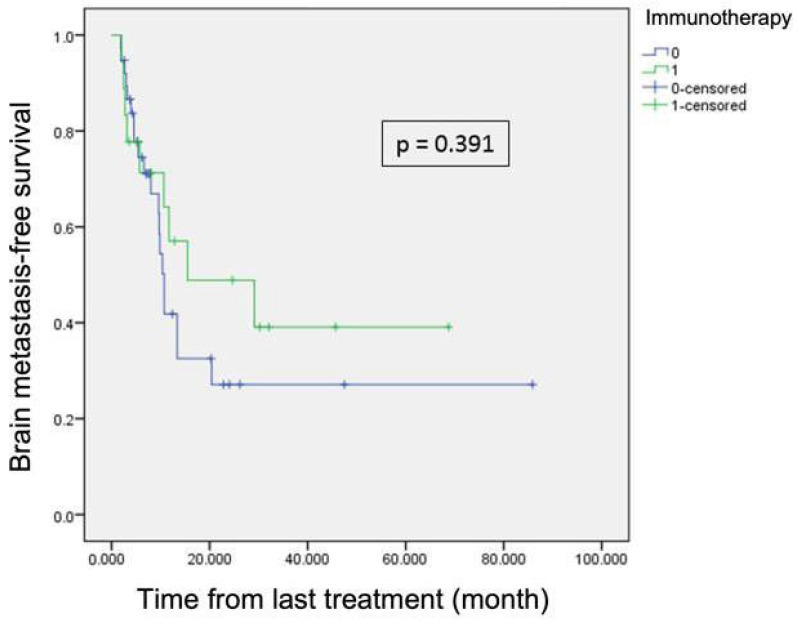
Brain metastasis-free survival for patients with immunotherapy in green and without in blue.

**Figure 3 cancers-16-04122-f003:**
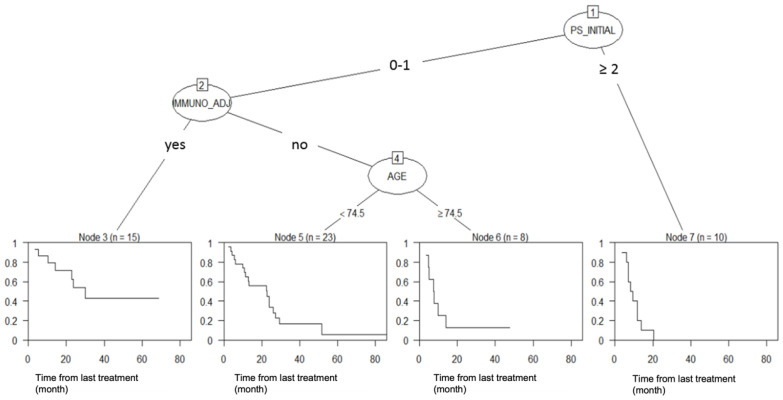
Recursive partitioning analysis of overall survival.

**Table 1 cancers-16-04122-t001:** Population characteristics.

	No PCI (N = 31)	PCI (N = 25)	*p*
Age mean (min–max)	64 (47–81)	65 (47–87)	0.656
Smoking status			0.580
- Non-smoker	1 (3.2%)	0	
- Former smoker	23 (74.2%)	15 (69.1%)	
- Current smoker	7 (22.6%)	9 (37.5%)	
Sex			0.580
- Men	18 (58.1%)	17 (68%)	
- Women	13 (41.9%)	8 (32%)	
Clinical tumor status			0.450
- T1	3(9.7%)	0	
- T2	6 (19.4%)	4 (17.4%)	
- T3	13 (41.9%)	9 (39.1%)	
- T4	9 (29%)	10 (43.5%)	
Clinical node status			0.112
- N0	3 (9.7%)	0	
- N1	3 (9.7%)	2 (8.3%)	
- N2	18 (58.1%)	10 (41.7%)	
- N3	7 (22.6%)	12 (50%)	
Clinical metastasis status			0.688
- M0	3 (9.7%)	4 (16%)	
- M1	28 (90.3%)	21 (84%)	
Stage			0.606
- Stage III	3 (9.7%)	4 (16%)	
- Stage IV	28 (90.3%)	21 (84%)	
Initial PS			0.622
- 0/1	25 (80.6%)	21 (84%)	
- 2	6 (19.3%)	4 (16%)	
Neurological history	6 (19.4%)	3 (33.3%)	0.716
Response befor PCI			0.155
- Stable	0	2 (8%)	
- Partial response	27 (87.1%)	22 (88%)	
- Complete response	4 (12.9%)	1 (4%)	
Immunotherapy	14 (45.2%)	4 (16%)	0.024
Initial MRI	15 (48.4%)	12 (48%)	0.977
Diagnostic year (mean)	2018	2015	0.000
Monocyte to lymphocyte ratio MLR (mean)	0.87	0.35	0.178
Neutrophil to lymphocyte ratio NLR (mean)	4.9	4.8	0.803

**Table 2 cancers-16-04122-t002:** Univariate and multivariate analysis of BMFS.

**Univariate Analysis**		
	HR (IC95%)	*p*-value
Age	0.99 (0.96–1.04)	0.820
Sex	0.99 (0.47–2.10)	0.998
Initial PS (0/1 vs. 2)	1.74 (0.64–4.72)	0.275
Neurological history	1.17 (0.44–3.07)	0.754
Response before PCI:		
- Partial response vs. stable	1.54 (0.21–11.44)	0.672
- Complete response vs. stable	1.24 (0.11–13.79)	0.858
Adjuvant immunotherapy	0.71 (0.32–1.57)	0.395
PCI	0.69 (0.33–1.47)	0.341
Hemoglobin	1.05 (0.86–1.29)	620
MLR < 0.12	1.17 (0.99–1.39)	0.069
**Multivariate Analysis**		
Adjuvant immunotherapy	0.41 (0.15–1.10)	0.077
PCI	0.52 (0.22–1.24)	0.140
MLR < 0.12	1.21 (1.01–1.45)	0.034

**Table 3 cancers-16-04122-t003:** Univariate and multivariate analysis of OS.

**Univariate Analysis**		
	HR (IC95%)	*p*-value
Age	1.03 (0.99–1.7)	0.057
Sex	1.7 (0.89–3.25)	0.106
Initial PS (0/1 vs. 2)	3.97 (1.59–9.89)	0.003
Neurological history	1.93 (0.92–4.06)	0.082
Response before PCI:		
- Partial response vs. stable	2.47 (0.34–18.07)	0.373
- Complete response vs. stable	1.37 (0.14–13.27)	0.785
Adjuvant immunotherapy	0.51 (0.25–1.03)	0.060
PCI	1.28 (0.70–2.33)	0.417
Hemoglobin	1.11 (0.92–1.09)	0.267
MLR < 0.12	1.05 (0.90–1.23)	0.515
**Multivariate Analysis**		
Age	1.01 (0.96–1.05)	0.780
Sex	1.18 (0.78–4.04)	0.675
Initial PS (0/1 vs. 2)	2.74 (1.23–6.13)	0.014
Neurological history	1.77 (0.79–4.05)	0.173
Adjuvant immunotherapy	0.59 (0.28–1.26)	0.177
PCI	1.20 (0.60–2.41)	0.601

## Data Availability

Research data are stored in an institutional repository and will be shared upon request to the corresponding author.

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
