# Peer review of "Lack of Prophylactic Cranial Irradiation for Extensive Small-Cell Lung Cancer in Real Life, with the Emergence of Immunotherapy"

_cancers, 2024, doi:10.3390/cancers16234122_

Round 1
Reviewer 1 Report
Comments and Suggestions for Authors
The authors have prepared a preliminary effort to address a long standing procedural issue of using prophylactic cranial irradiation for small cell carcinoma. In the era of immunotherapy it is clear that these therapies are valid, persistant and result in low metastasis.
Their study although small, sets the stage for review of other such procedures that are conducted in the field with little to no additive value. it is important to consider these and streamline therapy for cancer treatment so that the patients are left with minimal discomfort and better quality of life.
The authors must provide how immunotherapy per say is different from ther therapies in the treatment of SSC.
Minor comments :
In table 2 MLR<0.2 why is the p value a 0.0.082, is that relevant significance ? similar discrepencies are in table 3 and 4 for MLR<0.2. is this significant and they authors must delve into its significance
table 3 hemoglobin p value being 267 is that accurate?
They authors have performed RPA analysis but have failed to provide insight on how and why it is important in the results section
References: Reference 16 is incomplete
Author Response
Comments 1: [The authors must provide how immunotherapy per say is different from ther therapies in the treatment of SSC.]
Response 1: Thank you for this comment. In a goal to focus on the immunotherapy effect for SCLC, we removed and had some sentences and references in the introduction :
At line 59 "Furthermore, the emergence of immunotherapy against SCLC [18, 19] with controversial effects on BM raises questions about the role of PCI. Several studies about BM in NSCLC have reported intracranial activity signal, and significant improvement of intracerebral disease control rates with immunotherapy treatment [20 –22 ]."
Comments 2: [ In table 2 MLR<0.2 why is the p value a 0.0.082, is that relevant significance ? similar discrepencies are in table 3 and 4 for MLR<0.2. is this significant and they authors must delve into its significance]
Response 2: Thank you for pointing this out. We have corrected the tables 2 and 3. We have modified the discussion at line 204:
"In our study, we observed that the MLR ratio had a potential impact on BMFS. A lower MLR (<0.12) was associated with an increased risk of cerebral progression with an HR of 1.21 (1.01-1.45). These data are comparable with existing literature, particularly with the use of this ratio in nomograms attempting to predict the occurrence of BM in limited stage SCLC [27 ,28 ]. In contrast, other studies find a protective effect of a low MLR ratio in NSCLC [ 29,30 ] and also in all other histologies[31]. Further studies are needed to validate the real role of the MLR ratio. However, the use of a biomarker such as MLR could help to predict the risk of cerebral progression and then to select high-risk patients who could benefit from PCI,. "
Comments 3: [table 3 hemoglobin p value being 267 is that accurate? ]
Response 3: Thank you again for pointing this out, we have corrected the table 3. « 0.267 ».
Comments 4: [They authors have performed RPA analysis but have failed to provide insight on how and why it is important in the results section]
Response 4: We are agree with this comment and we thank you for this point. We have modified the discussion at line 187 :
" Furthermore, in our RPA results, immunotherapy stands out as the second most important factor affecting overall survival, following PS while PCI was not established as a relevant factor. These results support the impact of immunotherapy and underline the lack of PCI efficiency. "
Comments 5: [References: Reference 16 is incomplete]
Response 5: Thank you again for pointing this out. We have corrected, it's now reference 23
"Therneau TM, Atkinson EJ, An Introduction to Recursive Partitioning Using the RPART Routine. Mayo Clinic Fundation, rochester 1997 Available online: https://www.mayo.edu/research/documents/rpartminipdf/doc-10027257 (accessed on 20 November 2024)."
Reviewer 2 Report
Comments and Suggestions for Authors
The study by Dumas, A., et. al. in a cohort of patients (25) from the same institution shows the lack of improvement by prophylactic cranial irradiation in patients with lung cancer. This result is potentially useful for cancer therapy, I do recommend its publication.
Only a brief suggestion, the term “in immunotherapy era” looks odd and unnecessary in the title, the authors suggested in the text a future study on this subject (L209-210).
Author Response
Comments: [Only a brief suggestion, the term “in immunotherapy era” looks odd and unnecessary in the title, the authors suggested in the text a future study on this subject (L209-210).]
Response: Thank you for your review, we understand your comment but we wanted to emphasize the presence of immunotherapy, so we have proposed a new title: "Lack of prophylactic cranial irradiation for extensive small cell lung cancer in real life, with the emergence of immunotherapy”.
Reviewer 3 Report
Comments and Suggestions for Authors
Clinical results of this study showed no benefit of prophylactic cranial irradiation for patients with extensive small cell lung cancer. The findings were independent of whether the patients also received immunotherapy. This makes sense because it is unclear how prophylactic radiation could be helpful to any patient since this would only treat small foci of malignant cells but not prevent the development of new malignant cells.
Author Response
Comments : [Clinical results of this study showed no benefit of prophylactic cranial irradiation for patients with extensive small cell lung cancer. The findings were independent of whether the patients also received immunotherapy. This makes sense because it is unclear how prophylactic radiation could be helpful to any patient since this would only treat small foci of malignant cells but not prevent the development of new malignant cells.]
Response : We thank the reviewer 3 for his comment.
Round 2
Reviewer 3 Report
Comments and Suggestions for Authors
Clinical results of this study showed no benefit of prophylactic cranial irradiation for patients with extensive small cell lung cancer. The findings were independent of whether the patients also received immunotherapy. This makes sense because it is unclear how prophylactic radiation could be helpful to any patient since this would only treat small foci of malignant cells but not prevent the development of new malignant cells. However, some studies have reported that prophylactic cranial irradiation was associated with improved survival and reduced cumulative incidence of brain metastatic lesions in patients with small cell lung cancer.